# *Smilax weniae*, a New Species of Smilacaceae from Limestone Areas Bordering Guizhou and Guangxi, China

**DOI:** 10.3390/plants11081032

**Published:** 2022-04-11

**Authors:** Jie-Ying Feng, Xin-Jie Jin, Sheng-Lu Zhang, Jia-Wen Yang, Shi-Peng Fei, Yu-Song Huang, Yan Liu, Zhe-Chen Qi, Pan Li

**Affiliations:** 1Zhejiang Province Key Laboratory of Plant Secondary Metabolism and Regulation, College of Life Sciences and Medicine, Zhejiang Sci-Tech University, Hangzhou 310018, China; fjyfenne@163.com; 2College of Life and Environmental Sciences, Wenzhou University, Wenzhou 325035, China; xinjie_jin@yeah.net; 3Laboratory of Systematic & Evolutionary Botany and Biodiversity, College of Life Sciences, Zhejiang University, Hangzhou 310058, China; zslwantee@126.com; 4Guizhou Botanical Garden, Guiyang 550000, China; yangwen318@163.com; 5Guizhou Maolan National Natural Reserve, Libo 558400, China; A1078533135@126.com; 6Guangxi Key Laboratory of Functional Phytochemicals Research and Utilization, Guangxi Institute of Botany, Guangxi Zhuang Autonomous Region and the Chinese Academy of Sciences, Guilin 541006, China; huang-yusong@163.com (Y.-S.H.); gxibly@163.com (Y.L.)

**Keywords:** Old World, morphological trait, phylogeny, taxonomy

## Abstract

A new species, *Smilax weniae* (Smilacaceae), from Southwest China, is described and illustrated. The new species bears peltate leaves, which was previously a unique feature of *S**. luei*. However, it differs from the latter by having a broad ovate leaf blade, longer peduncle, and sexual dimorphic flowers. Further phylogenetic analyses revealed that the new species were placed in a unique position in a subclade of Old World *Smilax* based on ptDNA and nrITS sequences. Combining detailed morphological comparisons and molecular evidence, we validated that *S**. weniae* is an undescribed new species. Moreover, the plastome characteristics of *S. weniae* are reported.

## 1. Introduction

*Smilax* L., currently considered the sole genus of Smilacaceae [1], is one of the most abundant and easily recognizable climbing plants in various ecosystems of the Old World and the New World [2]. Typically, they are characterized by vines climbing or shrubs, being woody, less frequently suberect or herbs, dioecious and with tuberous or stoloniferous rhizomes, stems and branches which are usually prickly, paired petiolar tendrils, unisexual flowers with six tepals and either six fertile stamens or staminodes in the case of pistillate flowers [3]. Smilacaceae was split into four major clades by Qi et al. [1]: clade A: *Smilax aspera* L., clade B: mostly American species, clade C: prickless herbs, non-climbing species and *Heterosmilax*, mostly Asian species, and clade D: Asian–Australian–African woody species. Clade C and clade D are primarily from the Old World. Within clade C, sect. *Heterosmilax* is distinguished from other *Smilax* by their flowers with connate tepals [4]. The subclades in clade C, i.e., sect. *Heterosmilax*, sect. *Nervo-marginatae,* sect. *Vaginatae*, sect. *Ligneoriparia* and sect. *Nemexia* are all prickless. In addition, *Smilax* is a difficult genus to classify since its plants are dioecious and exhibit extensive phenotypic variations [5].

In 2020, we found a unique *Smilax* population with peltate leaves from Guizhou Manlan National Natural Reserve in Southwest China. The only formerly known *Smilax* species with peltate leaves is *Smilax luei* T. Koyama, which is endemic to Taiwan in Southeast China and can be easily distinguished by the morphological characteristics of the leaf blade, peduncle and male flower. In order to observe this special species, we conducted several follow-up fieldworks in 2021 and successfully collected its female & male flowers. During the subsequent herbarium work, we found two fruiting specimens (collected from Huanjiang County, Guangxi, and now deposited in the Herbarium of Guangxi Institute of Botany, IBK) representing this new species. To reveal its systematic position, this new species and 34 other representative Old World *Smilax* species were selected for phylogenetic studies. The unique morphology and systematic position both confirmed that it is an undescribed *Smilax* species, which is described and illustrated below. Additionally, the complete plastome of the novel species is reported.

## 2. Results

### 2.1. Morphological Comparison

Morphological comparisons between ‘*Smilax weniae*’ and *S. luei* T. Koyama are summarized in Table 1. The new species is distinctive in having sexually dimorphic flowers (male flower: tepals connate ca. 1/5; female flower: tepals separated, glabrous on both surfaces), longer peduncles, and ovate leaves.

### 2.2. Molecular Phylogenetic Analyses

The total alignment of the five plastid (pt) regions (*mat*K, *rbc*L, *ndh*A, *ndh*F, *rpl*16) was 7022 bp and included 97 variable sites, 49 of which were informative. The alignment of nuclear ribosomal (nr) ITS regions was 823 bp and included 95 variable sites, 46 of which were informative. Both phylogenetic trees show that *S. weniae* is included in clade C. The ptDNA ML cladogram (Figure 1) showed that *Smilax weniae* (PP = 0.95, ML BS = 99) is sister to a clade consisting of 20 *Smilax* species represented by sect. *Heterosmilax* + sect. *Nervo-marginatae* + sect. *Vaginatae* + sect. *Ligneoriparia*. While the nrITS ML cladogram (Figure 2) showed that *Smilax weniae* is sister to two species (*S. fui* and *S. ligneoriparia*) of sect. *Ligneoriparia* with weak support (PP = 0.74, ML BS = 90). The ML phylograms of ptDNA and nrITS (Appendix A) indicated that *S. weniae* had a deep genetic divergence with its sister clades. Occupying a unique systematic position, together with its distinctive morphological characteristics, *Smilax weniae* is verified to be a new species.

### 2.3. Taxonomic Treatment of the New Species

Taxonomic description of *Smilax weniae* P. Li, Z.C. Qi & Yan Liu., sp. nov. (Figure 3 and Figure 4).

Type: CHINA. Guizhou Province: Libo County, Weng’ang Town, Maolan National Natural Reserve, on the way from Jiuwei to Gengdushan, 835 m, 107°54′18.16″ E, 25°10′57.83″ N, 13 April 2021, Fl., *Zhechen Qi* Pan Li 009119 (holotype: ZM, isotypes: CSH, HZU, IBK, KUN, PE).

#### 2.3.1. Diagnosis

The new species resembles *Smilax luei* by bearing coriaceous peltate leaves with three primary and two marginal veins but differs in occasionally prickly stems (vs. prickless), ovate leaf (vs. lanceolate), and longer peduncle (2.5–10 cm vs. 1.5–2.5 cm).

#### 2.3.2. Additional Specimens Seen (Paratypes)

CHINA. Guizhou Province: Libo County, Weng’ang Town, Maolan National Natural Reserve, on the way from Jiuwei to Gengdushan, 795 m, 107°54′27.78″ E, 25°10′47.72″ N, 10 August 2020, *Pan Li, Meizhen Wang, Shenglu Zhang & Jinren Yu* WMZ333; *ibidem*, 957 m, 107°54′23.00″ E, 25°10′36.21″ N, 22 March 2021, *Pan Li, Zhechen Qi & Lianghai Yang* Pan Li 009014; *ibidem*, 792 m, 107°54′14.10″ E, 25°11′2.40″ N, flower in bud, 6 April 2021, *Zhechen Qi* Pan Li 009114; Guangxi Province: Huanjiang Maonan Autonomous County, Mulun Town, Xiayi, 880 m, Fr. (♀), 27 October 1991, *Dianqiangui Team* 70302 (IBK00199287!); Huanjiang Maonan Autonomous County, Mulun Natural Reserve, Xiazhai, 390 m, Fr. (♀), 4 October 2011, *Richeng Peng, Jing Liu & Chuanren Hu* ML0339 (IBK00312465!).

#### 2.3.3. Description

A perennial woody vine, climbing with tendrils. Stems subterete, glabrous, occasionally prickly. Petioles 1.2–3.5 cm long, narrowly winged for about 1/5 of their length, glabrous; abscission zone subapical; stipular tendrils, born on the basal portion of petioles, well-developed, grayish purple when young, then becoming brown at maturity. Leaf blade coriaceous, peltate, ovate, 5.5–14.5 cm long, 3.5–9.5 cm wide, with acuminate apex, base truncate, retuse or round, three primary and two marginal veins. The leaf is green and glossy above, glaucous beneath, purple or light yellow at youth, and most (not all) individuals have patchy white spots at maturity. Inflorescence of 1 umbel, born in the leaf axil, basally not prophyllate; peduncle 2.5–7.2(–10) cm long, straight, slightly compressed; umbel 10–28 flowered, hemispherical or spherical, base slightly thickened, globose, 3–4 mm in diameter. Pistillate flowers born on pedicels 4.0–7.0 mm long, red-yellowish; tepals: six, elliptic to ovate, 3.0–3.8 mm long, 1.9–2.1 mm wide, glabrous on both surfaces; staminodes: six; ovary superior. Staminate flowers born on pedicels 2.0–3.0 mm long, yellow–greenish; tepals: six, two whorls, 4–5 mm long, elliptic to oblong, adaxially involution, outer ones 3.0–4.0 mm wide, inner ones slightly narrower, outer ones and inner ones connate for ca. 1/5; stamens: six, the yellowish anthers on a short filament, two anther sacs obvious, adaxially involuted. Berries 1.0–3.0 mm in diameter, presumably red at maturity (based on IBK00199287!, IBK00312465!), Flowering in April, Fruiting in October.

#### 2.3.4. Etymology

The species is named in memory of Mrs. Hequn Wen, who was a researcher and vice director of Guangxi Institute of Botany from 1995 to 2000. Then she worked in the government of Liuzhou City, and tragically died in a car accident on a business trip to a remote location near the boundary of Guangxi and Guizhou. Wen was the first person to realize that *Smilax weniae* probably represents an undescribed species and provided the nude name ‘*Smilax peltatus* H.Q. Wen’ on an annotation label of the *Dianqiangui Team* 70302 (IBK00199287!) specimen, which was collected from Huanjiang County in Guangxi. The specific epithet she proposed clearly referred to the peltate leaves. Hence, we propose the Chinese name ‘盾叶菝葜 (dun ye ba qia)’.

#### 2.3.5. Distribution and Habitat

This new species is known from Libo County, southern Guizhou Province and Huanjiang Maonan Autonomous County, northern Guangxi Zhuang Autonomous Region, China. Till now, it has been recorded at one locality of Libo County and two localities of Huanjiang Maonan Autonomous County (Figure 5). It grows on the slopes of limestone mountains at 390–950 m.

#### 2.3.6. Conservation Status

*Smilax weniae* was found in Guizhou Maolan National Natural Reserve and Guangxi Mulun National Natural Reserve. Both reserves are primarily dedicated to the preservation of the subtropical karst forest ecosystem and uncommon wild animal and plant resources. The two reserves are quite abundant in plant diversity. We anticipate that more populations will probably be discovered in the near future. Nevertheless, the new species is currently known from three sites and has a restricted distribution area. More in field research is certainly needed for an appropriate definition of distribution, population size, locations (sensu IUCN), threats, etc., information which is necessary for the assessment of the conservation status according to the IUCN categories and criteria [6]. In fact, species with a restricted distribution could be attributed to different categories, as Critically endangered (CR) according to IUCN criteria B and C [7,8], as Vulnerable (VU) under IUCN criterion D [9,10], or as Data deficient (DD) [11], etc.

#### 2.3.7. Taxonomic and Evolutionary Relationships

Morphologically, the new species occasionally bears prickles on the stem, whereas the stems of other species in this clade are prickless [12]. In addition, the new species bears peltate leaves, which is a rare feature in the genus. Besides, in most known Smilacaceae species, the separation/connation patterns of male and female flowers tepals are consistent, i.e., separate in both genders, basally connate in both, or completely connate in both. The new species, however, shows an unusual sexual dimorphic pattern in which the tepals are basally connate in staminate flowers but separate in pistillate flowers. This pattern was first discovered in *Smilax hirtellicaulis* C.Y. Wu & C. Chen ex P. Li. which had closest phylogenetic relationship with sect. *Heterosmilax* [13]. Instead, *S. weniae* is sister to a clade consisting of representatives of four *Smilax* sections (including sect. *Vaginatae*, sect. *Ligneoriparia*, sect. *Heterosmilax* and sect. *Nervo-marginatae*) based on ptDNA loci (Figure 1, PP = 0.95, ML BS= %). Thus, this distinct characteristic (tepals are basally connate in staminate flowers but separate in pistillate flowers) is clearly a result of parallel evolution rather than a synapomorphy.

### 2.4. Characteristics of Plastome

The full length of the *S. weniae* plastome (GenBank Accession No. OL444944) is 158,204 bp and comprised of a large single copy region (LSC with 85,380 bp), a small single copy region (SSC with 18,467 bp), and two inverted repeat regions (IR with 27,179 bp). The overall GC content of the *S.*
*weniae* plastome is 37.2% and the GC content of the LSC, SSC and IR regions are 35.2%, 31.0%, and 42.4%. A total of 131 genes are included in the genome (85 protein-coding genes, eight rRNA genes, and 38 tRNA genes). Eighteen genes had two copies, which were comprised of seven PCG genes (*ndh*B, *rpl*2, *rpl*23, *rps*7, *rps*19, *ycf*1, ycf2), seven tRNA genes (*trn*V-GAC, *trn*R-ACG, *trn*N-GUU, *trn*L-CAA, *trn*I-GAU, *trn*H-GUG, *trn*A-UGC), and all four rRNA species (*rrn*16, *rrn*23, *rrn*4.5, *rrn*5). One gene had four copies, which is *trn*M-CAU gene. In the genome, nine protein-coding genes (*rps*16, *rpo*C1, *atp*F, *pet*B, *pet*D, *rpl*16, *rpl*2, *ndh*B, *ndh*A) had one intron, and *ycf*3, *clp*P, *rps*12 genes contained two introns (Figure 6).

## 3. Discussion

*Smilax weniae* is morphologically allied to *S. luei*, which is endemic to central Taiwan, China [14,15]. They both have distinct peltate leaves. Individuals with patchy white spots were found in both *S. weniae* and *S. luei*. They also share the same type of leaf vein character (3 primary and 2 marginal veins). However, *S. weniae* can be easily distinguished from *S. luei* by its unique male flower (tepals connate ca. 1/5), female flower (tepals glabrous on both surfaces), longer peduncle and ovate leaf blade (Figure 4). Furthermore, we found a few *S. weniae* individuals having prickles on their stems on occasion, while the stems of *S. luei* are always prickless. In addition, the peduncle length of the new species is usually longer than 2.5 cm, and the longest is even 10 cm, while the peduncle of *S. luei* is significantly shorter (1.5–2.5 cm). In addition to the above several morphological characteristics that can distinguish the new species from other species, it also has an unusual sexual dimorphic pattern, which is similar to *S. hirtellicaulis* (male flower: tepals connate ca. 1/5; female flower: tepals separated). *Smilax weniae* and *S. hirtellicaulis* both have similar broad ovate leaves and slightly compressed peduncles, *S. weniae* differs from the latter by having distinctive peltate leaves and prickly stems, and the female tepals are glabrous on both surfaces, while the female tepals of *S. hirtellicaulis* are thickened, and adaxially verruculose.

While incongruence was detected in the phylogenetic placement of *S. weniae* based on nrITS and ptDNA loci, the novel species status is confirmed. The ptDNA phylogeny provided more convincing statistical support for its sister relationship to four *Smilax* sections (Figure 1). The different placement inferred from nrITS could be simply due to a lack of information or a different evolutionary history between nuclear and plastid DNA. More extensive molecular systematic research is needed to elucidate the evolutionary history of *S. weniae* in the future. Based on morphological studies and phylogenetic analyses, *Smilax weniae* is confirmed to be a new species. We observed all suitable root apical materials for chromosome number, the number of chromosomes in *Smilax weniae* seems to be 60 (Appendix A), while the number of chromosomes in most *Smilax* species is 2*n* = 32, and a few species are 26, 28, 30, 60, 96. This means that *Smilax weniae* might be a polyploid. However, whether it is an autopolyploid or an allopolyploid remains unclear, and the exact number of chromosomes still needs further verification.

## 4. Materials and Methods

### 4.1. Morphological Observation

In 2020 and 2021, several field excursions were made to Gengdushan, Libo County, Guizhou, China to observe this species. Specimens with pistillate or staminate flowers were collected in 2021. At the same time, several main herbaria in China (HZU, IBK, NAS, PE, acronyms according to Thiers 2020) [16] were consulted to check if similar specimens could be located. Indeed, we found two additional specimens in IBK belonging to the new species, both of which have fruits (*Dianqiangui Team* 70302, IBK00199287!; *R.C. Peng* et al. ML0339, IBK00312465!). Based on our field observations, recent collections, and these historical specimens, we documented the morphology of this new species.

### 4.2. Molecular Methods and Phylogenetic Analyses

For phylogenetic analyses, we sampled one individual of the new species from Libo County, Guizhou Province, China, and thirty-four species of Smilacaceae including representatives of clade C and clade D (Table 2). *S**milax aspera* L. was set as an outgroup. All voucher specimens were deposited at the Herbarium of Zhejiang University (HZU) (Table 2).

Total genomic DNA was extracted from silica-dried tissue following a modified cetyltrimethylammonium bromide (CTAB) protocol [17]. The aqueous phase was extracted with 24:1 chloroform/isoamyl alcohol, and after isopropanol precipitation, the DNA was resuspended in Tris-ethylenediamine tetra-acetic acid (TE) buffer (pH 8.0). Based on their suitability to address inter-specific phylogenetic questions, five plastid DNA fragments (*ndh*A, *ndh*F, *rpl*16, *mat*K and *rbc*L) and a nuclear ribosomal ITS were employed for phylogenetic analyses. Amplification of the ITS region and *rpl*16 intron followed Cameron & Fu [2] and Fu & al. [18]. According to Shaw et al. [19], the PCR cycling conditions for the *ndh*A intron were 35 cycles of denaturation at 94 °C for 30 s, primer annealing at 55 °C for 30 s, and primer extension at 72 °C for 2 min. The primer design and amplification of *rbc*L and *mat*K refer to Qi et al. [1]. According to Qi et al. [1], the amplification of the plastid *mat*K gene was accomplished using designed primers ‘M3’: GCAACAATACTTCCTATATCCGCTTCT and ‘M4’: GAACTCTTCTAATAATCCCGAACCTAA. The PCR cycling conditions were template denaturation at 94 °C for 6 min prior to the start of PCR cycles, then amplified for 35 cycles of 1 min at 94 °C, 1.5 min at 53 °C, 2 min at 72 °C, and one final cycle of 12 min at 72 °C. Forward and reverse sequences were assembled using GENEIOUS v11.1.5 (Biomatters Ltd., Auckland, New Zealand). All sequences were deposited in GenBank (Table 2).

Phylogenetic analyses were performed using Bayesian inference (BI) and maximum likelihood (ML). BEAST v2.4.3 [20] was used to run the BI analyses. We conducted two independent runs of 100 million generations, with samples saved every 5000 generations. After assessing the results in Tracer v1.6 [21], we discarded the first 10% of the trees as burn-in. The log files were checked for convergence using Tracer. In both steps of our analyses, all ESS (explained sum of squares) values were well over 200; a maximum clade credibility tree was summarized with Tree Annotator v1.8.4 (included in the BEAST package). The ML analysis was performed using IQTREE v1.6.8 [22], of which the bootstrap values were calculated using 5000 replicates with the best selected TPM3u+F+R2 model. Trees were visualized using FigTree v1.4.3 [23].

### 4.3. Plastome Sequencing and Analysis of Smilax weniae

Whole plastome sequences were generated using the Illumina HiSeq-2500 platform (Illumina Inc., San Diego, CA, USA). In total, about 22.49 million high-quality clean reads (150 bp PE read length) were generated with adaptors trimmed. Aligning, assembly, and annotation were conducted by GetOrganelle v1.7.0c [24], MAFFT [25], GeSeq [26] and GENEIOUS v11.0.5. The circular structure of ptDNA was drawn by Organellar Genome DRAW [27].

### 4.4. Chromosome Counts

Chromosomal Counts were obtained by the root tip squash methods as described in Kong et al. [12]. The vigorously growing root tips were treated with 0.05% colchicine solution for 4 h, fixed with Carnot’s fixative solution (glacial acetic acid: absolute ethanol = 1:3) for 24 h, and stored at 4 °C with 70% ethanol for later use. During tableting, the root tips were washed with distilled water, dissociated with 1 mol/L hydrochloric acid in a constant temperature water bath at 60 °C for 50 s, rinsed with distilled water, and then drip-dyed with a modified phenolic fuchsin solution (Carbol-Fuchsin) for tableting. Chromosome counts were observed for at least 30 complete metaphases. Chromosome spreads were observed using 100× light microscopy.

## Figures and Tables

**Figure 1 plants-11-01032-f001:**
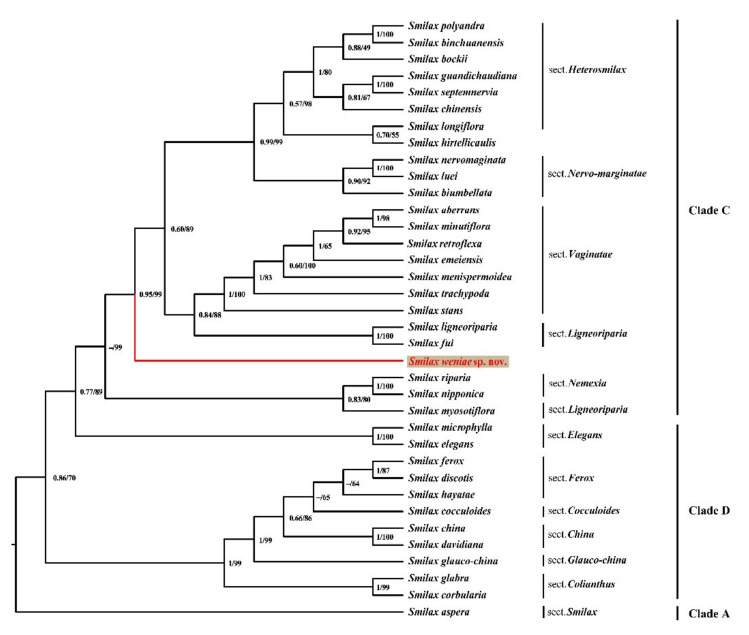
Maximum likelihood cladogram of the ptDNA of *Smilax weniae* and 34 other *Smilax* species, representing an Old World Smilacaceae clade. Posterior probabilities (PP > 0.50) and bootstrap values (BS > 50%) based on Bayesian and maximum likelihood (ML) analyses are shown near the branches. Clade information adopted from Qi et al. [1].

**Figure 2 plants-11-01032-f002:**
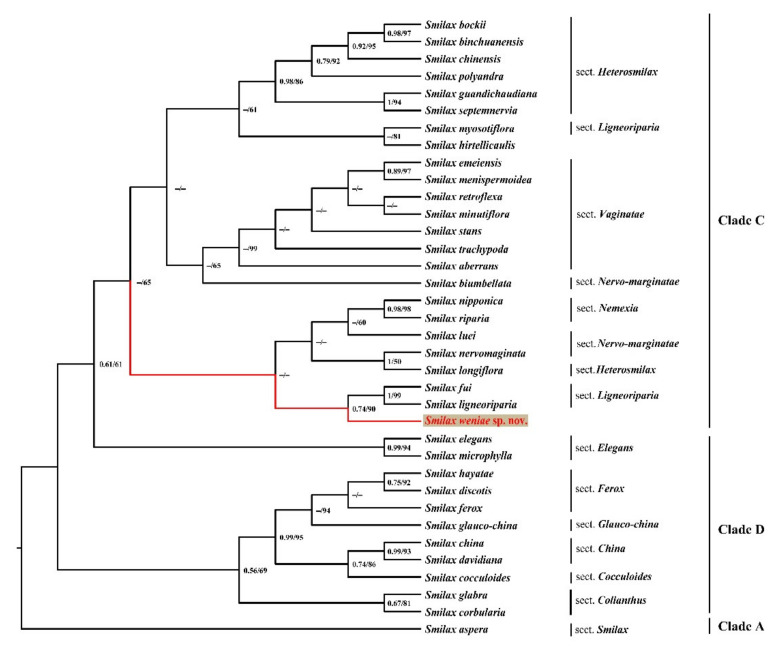
Maximum likelihood cladogram of the nrITS of *Smilax weniae* and 34 other *Smilax* species, representing an Old World Smilacaceae clade. Posterior probabilities (PP > 0.50) and bootstrap values (BS > 50%) based on Bayesian and maximum likelihood (ML) analyses are shown near the branches. Clade information adopted from Qi et al. [1].

**Figure 3 plants-11-01032-f003:**
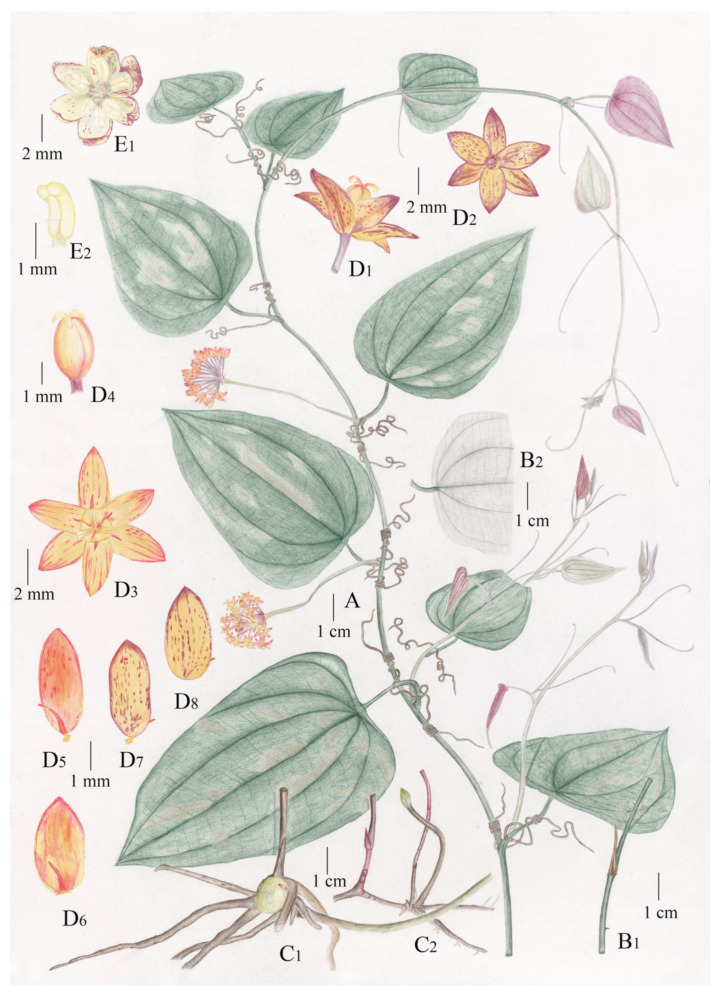
Illustration of *Smilax weniae* P. Li, Z.C. Qi & Yan Liu, sp. nov. (A) Flowering plant (♀); (B1) Stem, with prickles occasionally; (B2) Leaf in abaxial view; (C1) Rhizome; (C2) Rhizome with bud; (D1) Pistillate flower in side view; (D2) Pistillate flower in rear view; (D3) Pistillate flower in front view; (D4) Ovary and stigma; (D5) Inner tepal (♀) in front view, with a staminode; (D6) Outer tepal (♀) in front view, with a staminode; (D7) Inner tepal (♀) in rear view; (D8) Outer tepal (♀) in rear view; (E1) Staminate flower in front view; (E2) Stamen. Drawn by Xin-Jie Jin.

**Figure 4 plants-11-01032-f004:**
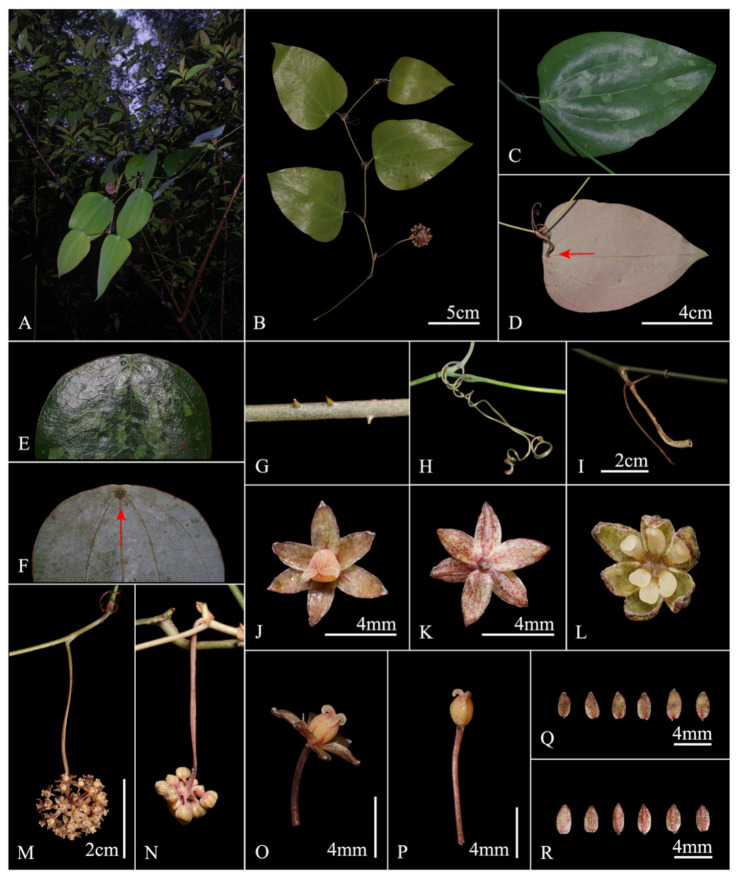
*Smilax weniae* P. Li, Z.C. Qi & Yan Liu, sp. nov. (**A**) Plant habit; (**B**) Flowering branch; (**C**) Stem with a leaf in adaxial view; (**D**) Stem with a leaf in abaxial view; (**E**) Leaf base in adaxial view; (**F**) Leaf base in abaxial view, showing peltate leaf; (**G**) Stem, occasionally prickly; (**H**) Tendrils; (**I**) Petiole; (**J**) Pistillate flower in front view; (**K**) Pistillate flower in rear view; (**L**) Staminate flower in front view; (**M**) Pistillate inflorescence; (**N**) Staminate inflorescence, flowers unopened; (**O**) Pistillate flower in side view; (**P**) Pistillate pedicel and pistil in side view; (**Q**) Tepals in front view; (**R**) Tepals in rear view. The red arrow points to the petiole insertion point of the peltate leaf.

**Figure 5 plants-11-01032-f005:**
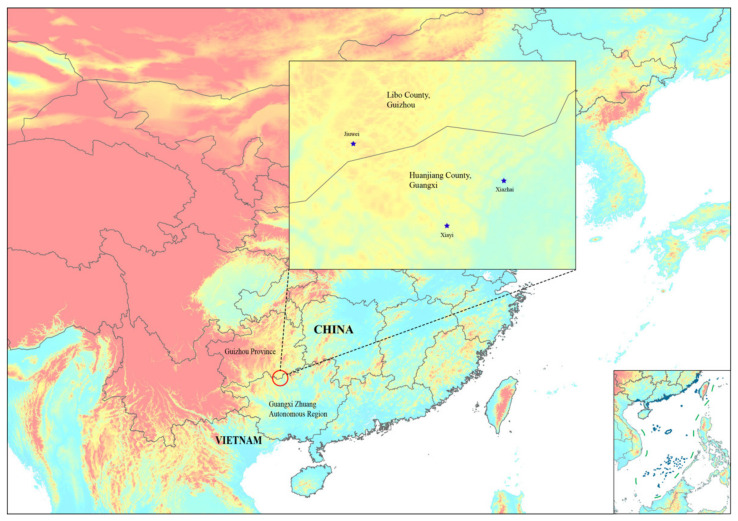
Distribution records of *Smilax weniae* P. Li, Z.C. Qi & Yan Liu. sp. nov. (blue stars) from Libo County, Guizhou Province, and Huanjiang County, Guangxi Zhuang Autonomous Region, China.

**Figure 6 plants-11-01032-f006:**
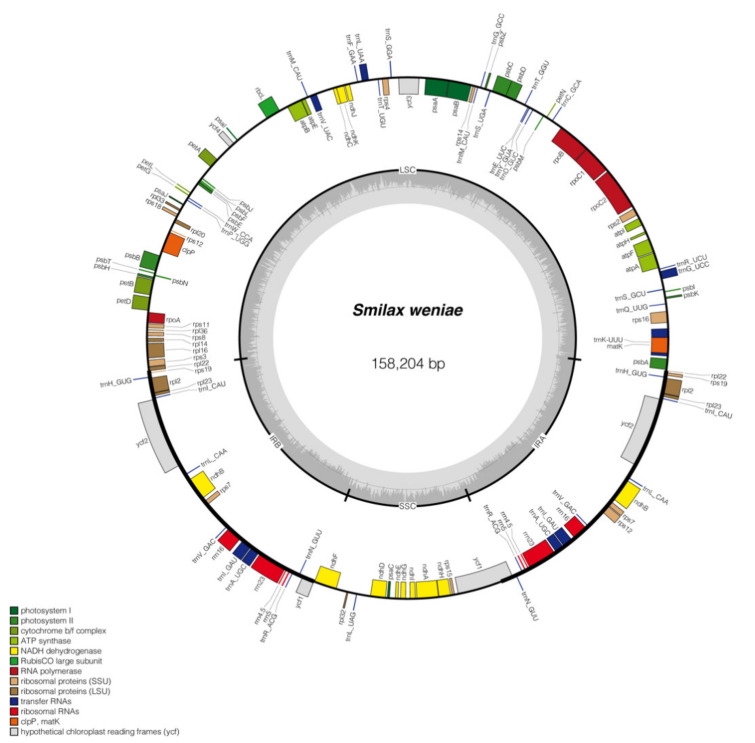
Plastome map of *Smilax weniae*. The inner dark gray circle corresponds to GC content and the inner light gray circle corresponds to the AT content. Different colors are used as a representation of distinctive genes within separate functional groups.

**Table 1 plants-11-01032-t001:** Morphological comparison between *Smilax weniae* and *Smilax luei* T. Koyama.

Characters	*S. weniae*	*S. luei*
Stem	prickless or occasionally prickly	prickless
Leaf blade	ovate, peltate, leaf base truncate, retuse or round, 5.5–14.5 cm long, 3.5–9.5 cm wide	lanceolate, peltate, leaf base retuse or round, 3–13 cm long, 1–3.5 cm wide
Male flower	tepals connate ca. 1/5, stamens 6	tepals separated completely, stamens 9 (rarely 10),
Female flower	tepals glabrous on both surfaces	tepals minutely verruculose on the outer surface
Peduncle	2.5–7.2(–10) cm long	1.5–2.5 cm long

**Table 2 plants-11-01032-t002:** Taxa included in the present study with voucher information and GenBank accession numbers for the sequences of internal transcribed spacer (ITS) of nuclear ribosomal DNA, *mat*K, *rbc*L, *ndh*A, *ndh*F and *rpl*16. Missing sequences are indicated by a dash (-). Accession numbers in bold are newly generated sequences.

Family	Species	Voucher	ITS	*mat*K	*rbc*L	*ndh*A	*ndh*F	*rpl*16
Smilacaceae	*Smilax weniae* P. Li, Z.C. Qi & Yan Liu. sp. nov.	P. Li 009119 (HZU)	**OL677459**	**OL504525**	**OL504528**	**OL504526**	**OL504527**	**OL504529**
	*S*. *aberrans* Gagnep.	C. Fu 20037 (HZU)	JF461346	JF461377	MT105074	KF818408	KF818438	JF461422
	*S. aspera* L.	BQ 0908304 (HZU)	JF461347	JF461378	MT105075	KF818409	KF818439	KC511399
	*S. biumbellata* T. Koyama	C. Fu 20022 (HZU)	JF461351	JF461383	MT105084	MT104985	MT104889	JF461427
	*S. china* L.	C. Fu 20006 (HZU)	JF978671	JF956367	JF944328	KC522265	KC5213150	KC511414
	*S. cocculoides* Warb. ex Diels	C. Fu 0903122-1 (HZU)	JF978680	JF956376	JF944337	KF818413	KF818443	KC511416
	*S. corbularia* Kunth	J. Li 0024298 (KRIBB)	KC511496	KC511356	MT105091	MT104994	MT104898	KC511417
	*S. davidiana* A. DC.	C. Fu Fw 108 (HZU)	KC511498	KC511358	MT105093	KF818414	KF818444	KC511421
	*S. discotis* Warb.	C. Fu Fw 111 (HZU)	JF956388	KC511359	JF944349	KF818415	KF818445	KC511422
	*S. elegans* Wall.	P. Li 0908300-1 (HZU)	JF978692	JF956392	JF944353	KC522275	KC5213160	KC511423
	*S. emeiensis* J.M. Xu	0610004 (HZU)	JF461353	JF461385	MT105097	MT105000	MT104904	JF461429
	*S. ferox* Wall. ex Kunth	C. Fu 20059 (HZU)	JF978696	JF956396	JF944357	KF818417	KF818447	KC511426
	*S. fui* Z.C. Qi & P. Li	C. Fu 10043 (HZU)	MT104825	MT104854	MT105150	MT105047	MT104951	MT105173
	*S. glabra* Roxb.	C. Fu 903107 (HZU)	JF978699	JF956399	JF944360	KF818418	KF818448	KC511427
	*S. glauco-china* Warb. ex Diels	C. Fu Fw114 (HZU)	JF978706	JF956406	JF944367	KF818420	KF818450	KC511429
	*S. hayatae* Koyama	C. Fu 0903151 (HZU)	MT104828	MT104857	MT105103	KF818422	KF818452	MT105176
	*S. hirtellicaulis* C.Y. Wu & C. Chen ex P. Li.	XJJin.HK01 (HZU)	KX712229	KX712230	KX712231	-	-	-
	*S. ligneoriparia* C.X. Fu & P. Li	P. Li 0904082 (HZU)	JF461359	JF461395	MT105118	MT105015	MT104919	JF461438
	*S. luei* T.Koyama	X. Liu 0809016 (HZU)	KC511503	JF461396	MT105120	MT105017	MT104921	KC511440
	*S. microphylla* C.H. Wright	C. Fu 20007 (HZU)	JF978747	JF956443	JF944407	KC522308	KC5213193	KC511445
	*S. menispermoidea* A. DC.	L. Gao 08898 (KUN)	JF461360	JF461397	MT105121	MT105020	MT104924	JF461439
	*S. myosotiflora* A. DC.	C. Fu 09013 (HZU)	KC511505	KC511366	MT105126	MT105024	MT104928	KC511446
	*S. nervo-marginata* Hayata	C. Fu 2010422 (HZU)	JF461363	JF461400	MT105130	KF818424	KF818454	KC511451
	*S. nipponica* Miq.	C. Fu 950174 (HZU)	AY775244	JF461402	MT105132	KF818425	KF818455	JF461443
	*S. riparia* A. DC.	C. Fu 912691 (HZU)	AY775234	JF461407	MT105142	KC522335	KC5213220	AY775217
	*S. retroflexa* (F.T. Wang & Tang) S.C. Chen	P. Li 0904056 (HZU)	JF461366	JF461406	MT105141	MT105041	MT104945	JF461447
	*S. stans* Maxim.	Y. Wan 0811021 (HZU)	JF461368	JF461413	MT105157	MT105052	MT104956	JF461449
	*S. minutiflora* F.T. Wang	C. Fu 20013 (HZU)	AY775256	JF461416	MT105163	MT105057	MT104961	AY775229
	*S. trachypoda* J.B. Norton	P. Li 0907257 (HZU)	JF461370	JF461415	MT105162	MT105055	MT104959	JF461451
	*S. chinensis* (F.T. Wang) P. Li & C.X. Fu	C. Fu20011 (HZU)	JF461342	JF461372	MT105064	KF818400	KF818430	JF461418
	*S. gaudichaudiana* Kunth.	BQ 0902015 (HZU)	KX394645	KX432982	KX394669	MT104873	MT104968	JF461419
	*S. bockii* Warb. ex Diels	C. Fu 010807 (HZU)	AY775257	JF461374	MT105066	KF818401	KF818431	AY775230
	*S. longiflora* (K.Y. Guan & Noltie) P. Li & C.X. Fu	C. Fu 9908 (HZU)	JF461344	JF461375	MT105067	MT104969	MT104874	JF461420
	*S. polyandra* (F. Gagnep.) P. Li & C.X. Fu	P. Li 0905145 (HZU)	KC511480	KC511340	MT105068	MT104970	MT104875	KC511389
	*S. septemnervia* (F.T. Wang & Tang) P. Li & C.X. Fu	P. Li 0908310 (HZU)	MT104844	MT104867	MT105148	MT104971	MT104876	MT105180
	*S. binchuanensis* P. Li & C.X. Fu	C. Fu 20019 (HZU)	JF976617	JF461376	JF941922	MT104877	MT104972	JF461451

## Data Availability

The molecular data that support the findings of this study are openly available in GenBank (see Table 2).

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
