# Peer review of "Smilax weniae, a New Species of Smilacaceae from Limestone Areas Bordering Guizhou and Guangxi, China"

_plants, 2022, doi:10.3390/plants11081032_

Round 1
Reviewer 1 Report
The paper aims to publish a new Smilax species with peltate leaves, of which only one other species of this genus has this character state. The morphological distinctions between these two peltate-leaved species are stated. In addition, a molecular study indicates this new species has its peculiar phylogenetic position, thus supporting the specific status of this new species.
This paper is outstanding and almost needs no revision. However, the authors may want to clarify the English name of the Chinese geographical locality. For example, the name of the natural reserve on line 119 is Maolan, but the reviewer suspect “Mulun” is the correct one as this locality is in Guangxi. There is no herbarium label information in the manuscript, and the spelling “Maolan” is possibly still accurate as the collectors might spell that way. Please check all the English geographical names, and make corrections or explanations.
The discussion of taxonomic and evolutionary relationships has some unclear points. First, if the species Smilax hirtellicaulis is essential in the relationship study, why is this species not included in the molecular research? Especially some of the authors must have this species material in hand. If possible, the inclusion of Smilax hirtellicaulis in the molecular phylogeny should bear more strength of the taxonomic and evolutionary discussions. Second, is the branch length in Fig. 1 the true reflection of the difference of branch length? If not, it only indicates that this new species is very close to Clase C, and outgroup species are far from this new species. This new species is way outside of the Clade C, and it might be included in other Clades if other Smilax species were included in the study. Although the molecular survey in this paper does not fully consider all clustering principles, it still indicates this new species is distinct, and the specific status is warranted.
Reviewer 2 Report
The results of morphological and molecular analyses allowed the authors to validate that Smilax weniae is a new species living in a restricted area in Southwest China. However, more precision is needed in the presentation of the results.
First of all, authors should state how many samples they have collected and studied. They should also list all the morphological traits analyzed, perform an appropriate statistical analysis, and show the statistical results at least for the characters differing from S. luei (it is not enough to show only the range of variability).
Molecular analyses clearly showed that the new species is sister to a clade of 22 Smilax species. In this case only one sample has been analyzed. I think that when a new species is described, a survey of the molecular variability within the discovered population is needed. More samples should be studied also from this point of view.
Moreover, in my opinion, an exhaustive description of a new species cannot ignore the analysis of the chromosome number and studies about its reproduction mode. I strongly recommend adding this information, which is completely absent from the manuscript.
The section of Materials and Methods needs to be improved. Primers and methodology of amplification should be extensively described. Please, provide information about softwares used to obtain ML dendrogram.
Reviewer 3 Report
The manuscript deals with the description of a new species of Smilax. The topic is interesting and inside the scope of the journal. The English is ok. The results and discussion are somewhat essential, but clearly presented. The Discussion (and Conclusion?) is very short. It must be implemented, for example with more comparison with other similar species or similar cases.
I am not sure if the Journal Plants needs Material & Methods paragraph after Discussion. I think this paragraph is before Results. Check it. If so, the authors have to change the numbering of the paragraphs (and subparagraphs) and also of the cited literature (for ex. Thiers 2020 not number 10 bur number 6).
I have one relevant concern. The new species and most of the manuscript are based on the differences between the new species and S. luei, as this latter was until now the only Smilax species with peltate leaves, as stated by the authors. However, looking to Figs. 2-3, the new species, S. weniae, seems not to have peltate leaves! The molecular analyses confirmed that S. weniae is a new species, but if the leaves of the new species are not peltate, the morphological analysis (in particular the diagnosis) must be re-written. The authors have to answer this important point.
Therefore, I suggest major revision.
Minor points:
Line 26: Delete “Smilax weniae” from the Keywords. Do not repeat in the keywords words already included in the Title
Line 56: “2.1. Morphological comparison” instead of “2.1. Morphology comparison”
Line 57: “comparison” instead of “comparisons”
Line 57: “is” instead of “are”
Line 61: “comparison” instead of “comparisons”
Line 61: “between Smilax weniae and” instead of “between ‘Smilax weniae’ and”
Line 77: “that Smilax weniae (PP = 1.0” instead of “that the ‘Smilax weniae’ (PP = 1.0”
Line 79: “characteristics, S. weniae is verified” instead of “characteristics, ‘S. weniae’ is verified”
Lines 161-166: The “Conservation Status” subparagraph needs improvements. Many of the new described species today are accompanied by a first evaluation of their conservation status according to IUCN protocol. Therefore, I suggest adding the following sentences at the end of the subparagraph (line 166):
Nevertheless, the new species is currently known from three sites and has a restricted distribution area. More in field research is certainly needed for an appropriate definition of distribution, population size, locations (sensu IUCN), threats, etc., information which is necessary for the assessment of the conservation status according to the IUCN categories and criteria [6]. In fact, species with a restricted distribution could be attributed to different categories, as Critically endangered (CR) according to IUCN criteria B and C [7,8], or as Vulnerable (VU) under IUCN criterion D [9,10], or as Data deficient (DD) [11], etc.
[6] IUCN (2022) Guidelines for using the IUCN Red List categories and criteria. Version 15. Prepared by the Standards and Petitions Committee. Available from: https://www.iucnredlist.org/resources/redlistguidelines (accessed 19 February 2022).
[7] Wagensommer, R.P.; Bartolucci, F.; Fiorentino, M.; Licht, W.; Peccenini, S.; Perrino, E.V.; Venanzoni, R. First record for the flora of Italy and lectotypification of the name Linum elegans (Linaceae). Phytotaxa 2017, 296 (2), 161–170. https://doi.org/10.11646/phytotaxa.296.2.5
[8] Wagensommer, R.P.; Venanzoni, R. Geranium lucarinii sp. nov. and re-evaluation of G. kikianum (Geraniaceae). Phytotaxa 2021, 489 (3), 252–262. https://doi.org/10.11646/phytotaxa.489.3.2
[9] Blasco, F.A.; Rubite, R.R.; Cortes, J.C.; Alejandro, G. J. D. Begonia lanuzaensis (sect. Petermannia, Begoniaceae) a new species from Surigao del Sur, Mindanao Island, Philippines. Phytotaxa 2021, 523 (3), 203–207. https://doi.org/10.11646/phytotaxa.523.3.1
[10] Swanepoel, W.; Cauwer, V. de; Van Wyk, A. E. A new rheophytic species of Syzygium (Myrtaceae) from the lower Kunene River of Angola and Namibia. Phytotaxa 2021, 491 (4), 281–290. https://doi.org/10.11646/phytotaxa.491.4.3
[11] Chinchilla, I. F. A new tree species of Cupania (Sapindoideae, Sapindaceae) from Quepos, Costa Rica. Phytotaxa 2020, 475 (3), 178–186. https://doi.org/10.11646/phytotaxa.475.3.2
Line 166: “will probably be discovered” instead of “will be discovered”
Line 175: “Smilax hirtellicaulis”: add author(s) of the species
Round 2
Reviewer 2 Report
None of my suggestions have been taken into consideration. The inserted text contain new information which should be better integrated. There are several typos.
Reviewer 3 Report
The authors followed the suggestions, answered my questions, corrected mistakes
and improved the manuscript.
Author Response
Reviewer3: The authors followed the suggestions, answered my questions, corrected mistakes and improved the manuscript.
Response: Thanks for your effort. We have carefully checked the manuscript again to correct any mistakes.
Round 3
Reviewer 2 Report
In my opinion, it would have been appropriate to publish and discuss the chromosome number. However, I thinke the results showed are sufficient to support the author's thesis.
Author Response
We accepted your suggestion to add and publish chromosome number of new species in Discussion and added chromosome map in supplementary material (Figure S3). Add the materials and methods of chromosome counts on line 397.
